# Occupation and Sickness Absence in the Different Autonomous Communities of Spain

**DOI:** 10.3390/ijerph182111453

**Published:** 2021-10-30

**Authors:** Matilde Leonor Alba-Jurado, María José Aguado-Benedí, Noelia Moreno-Morales, Maria Teresa Labajos-Manzanares, Rocío Martín-Valero

**Affiliations:** 1Medical Unit, National Institute of Social Security, 29010 Malaga, Spain; matilde-leonor.alba@seg-social.es; 2National Institute of Social Security, 28036 Madrid, Spain; mjbenedi@yahoo.es; 3Department of Physiotherapy, Faculty of Health Science, Ampliacion de Campus de Teatinos, University of Malaga, C/Arquitecto Francisco Peñalosa 3, 29071 Malaga, Spain; nmm@uma.es (N.M.-M.); mtlabajos@uma.es (M.T.L.-M.)

**Keywords:** sickness absence, occupation health, occupational class, Autonomous Community

## Abstract

The occupation of a worker is a determining factor of sickness absence (SA) and can influence both the beginning and continuation of the latter. This study describes SA in Spain, separately in the different Autonomous Communities (AC) in relation to the occupation of workers, with the aim of determining the possible differences in its frequency and duration, relating it also to the diagnosis. A total of 6,543,307 workers, aged 16 years and older, who had at least one episode of SA in the year 2019, constituted the study sample. The obtained results indicate that SA is more frequent and shorter in more elemental occupations. The average duration increases with age and is longer in women, except in technical and administrative occupations, where there is no gender divide. Sickness absences caused by musculoskeletal and mental disorders are more frequent in the lower occupational classes, although their average duration is shorter than in other, more qualified groups. The ACs with shorter duration in almost all the occupational groups are Madrid, Navarre and the Basque Country. In conclusion, SA is more frequent and shorter in lower occupational classes.

## 1. Introduction

In Spain, sickness absence (SA) is defined as that situation in which a worker cannot perform his/her job due to illness or injury and receives financial aid from the Social Security system [1].

The maximum duration of this financial aid is one year, which can be extended up to 6 months if it can be predicted that the worker may improve his/her health state in that period and go back to work. In order to be entitled to this benefit, in case of common disease, the worker must have contributed a minimum of 180 days in the previous 5 years at the beginning of the SA. In the case of professional illness or accident, whether occupational or not, no previous contribution period is required [1].

This situation affects both the workers who carry out their job as employees and those workers who work for themselves, i.e., self-employed workers or business owners.

The recognition of SA requires a medical certificate, generally from a doctor of the National Health System in the case of common eventualities, or a doctor of one of the Insurance Companies Linked to the Social Security System in the case of professional eventualities. These doctors are also in charge of issuing certificates for the worker if the latter continues to be sick or injured, declaring that he/she cannot go back to work, until he/she improves to the point that he/she can resume his/her working activity. In some cases, the consequences of the disease or accident are so severe that the worker can be permanently unable to perform his/her job or any job in general; this recognition is done by the Disability Assessment Team, which proposes a permanent disability to the National Institute of Social Security (INSS) [2,3].

Like most European countries, Spain’s public sickness insurance spending is a major component of its social security system. The total financial cost of SA in consolidated Social Security budgets for 2019 in Spain reached 11,554,711.16 €, which is 8.89% of the general Social Security budget [4]. Besides, the importance of SA caused by disease or accident is not only determined by the social and financial costs that each State dedicates to it, but also by the resources that employers assign to it, as well as the productivity losses and the deterioration of these workers’ health. It has been reported that work absenteeism due to illness is, in itself, a risk factor for falling sick, new periods of SA, unemployment, permanent disability pension, social exclusion and death [5,6,7,8]. Long SA periods generate both an early retirement from the job market, a slower salary increase and an impoverishment of household wealth [9].

Knowledge of the determinants of sickness absence and of the accumulation of sickness absence days is important to be able to target effective measures that aim to prevent both incidence and prolongation of sickness absence and to curb its harmful consequences [10].

Both at the beginning and continuation (i.e., duration) of this situation, there are multiple influencing factors, especially those related to the health-disease binomial, as well as individual, job and socioeconomic factors [11].

Socioeconomic factors are of great relevance regarding SA, since their influence on the health of workers has been demonstrated in numerous studies throughout history [11,12,13].

Among the socioeconomic factors, the more important ones in relation to SA are job hierarchy or occupational class, education level, income, unemployment and other economic aspects [12,14,15].

In all countries, it is stated that SA is higher in the lower occupational classes, that is, those in which workers have a greater physical load and less intellectual responsibility [16]. However, studies indicate that this fact occurs in all occupational classes and remains throughout the years, affecting both men and women. This is also observed in non-manual workers whose activity is carried out in a low occupational class, with great physical and mental demands, as in the case of nurses [17].

Regarding the above mentioned, some studies have reported an association between the occupational class and the duration of SA. It has been observed that, in blue collar workers, SA is shorter and more frequently caused by musculoskeletal problems than in white collar workers. However, when a white collar worker initiates a SA by musculoskeletal cause, it is usually a long SA (longer than 6 months), more frequently associated with mental problems, and with the worker being more likely to be permanently disabled [18]. The authors justify this assertion by considering that, when white collar workers initiate a SA by musculoskeletal cause, this disease is usually more severe and, since many cases are associated with psychic problems, the prognosis worsens, and the duration of the SA and the probability of developing a permanent disability increase.

The studies conducted by Pekkala et al. are also along these lines, indicating that SA is much more frequent in the lower occupational classes, in both men and women [17].

This has been recently corroborated by Blomgren and Jäppinen (2021), who included all occupational classes (employees, business owners or self-employed workers, and unemployed individuals). That is, SA is more frequent in the lower manual occupational classes, regardless of whether they are self-employed workers or employees. In the study mentioned, SA from musculoskeletal problems was similar in the group of employees than in the group of self-employed workers. However, SA by mental problems was more frequent in employees than in business owners. Unemployed individuals also showed a greater SA rate and duration than expected for mental health problems. This seems to be due to the fact that the lower occupational classes have worse physical working conditions and greater psychosocial risks, as they are under more compulsion and have a lower decision-making capacity in their job position. In the group of business owners, the authors of the mentioned study state that both men and women have better mental health due to the natural selection of healthier and psychologically stronger people to occupy these positions, along with the positive influence of their high decision-making capacity and autonomy in their job position [10].

Other studies, such as that of Piha et al., attempted to find a correlation between several socioeconomic factors, mainly education, occupational class, income and SA. These authors found that higher education levels, occupation classes and income were strongly related to a lower SA rate. Controlling for these three variables, education level and occupational class showed stronger correlations with low levels of SA compared to income [19].

Spain is divided into 19 Autonomous Communities. In each region there are significant differences in education, economic situation, unemployment rate and public health system. By sex, the percentage of men and women is balanced. The number of women is slightly higher, although the number of working men remains higher. By age, the regions with the youngest population are those on the Mediterranean coast and also Castile La Mancha, Extremadura and the Community of Madrid. The northwestern regions are more aged. The regions with the highest rates of active population were the Canary Islands, the Balearic Islands, the Community of Madrid and Catalonia and those with the lowest rates were the Cantabrian, Western and Central regions. The service sector contained the majority of the work-force (76%) in the islands, the Community of Madrid and Andalusia. Industry was the predominant sector in Navarre, the Basque Country and La Rioja; construction in Castile La Mancha; and agriculture and farming in Murcia [20].

A descriptive study of SA was conducted in Spain in 2018, and also by Autonomous Community (AC), although SA was not analysed as a function of the worker’s occupation [4]. The authors concluded that further research was necessary due to the differences found in SA in the different ACs regarding age, gender, whether the worker was employed or self-employed, the cause of the sick leave (work-related or non-work-related SA) and the relationship with the activity in the company.

Therefore, the aim of the present study was to analyze the differences in SA in terms of occupational class, according to the International Standard Classification of Occupations (ISCO-08) [21], focusing on the different regions of the country and according to the main characteristics of SA and differences between AC. Thanks to this study, new measures could be introduced to improve workplaces, workers could take up their jobs sooner, SA benefits could be managed more efficiently in AC, and the negative consequences of SA could be reduced.

## 2. Materials and Methods

### 2.1. Study Design and Participants

A population study was carried out using the databases of the INSS (https://www.seg-social.es/wps/portal/wss/internet/EstadisticasPresupuestosEstudios/Estadisticas/EST45/EST46 (accessed on 25 February 2021)) and the Spanish Statistics Institute (INE) (https://www.ine.es (accessed on 25 February 2021)) including 6,543,307 SA processes (all regimes, including self-employed workers) from the year 2019.

This is an observational study. The statistical calculations used to refer to all the people affiliated with the system who are entitled to receive the SA benefit, calculated according to the data provided by the Public Employment Service, the General Treasury of Social Security, INSS and the Social Institute of the Navy.

The study population was constituted by workers aged 16 years and older who contributed to the Social Security system, excluding civil workers of the State Civil Servant Mutual Society, the General Judicial Mutual Society and the Armed Forces, since their SA management and control system is different from that of the rest of workers.

We analysed a total of 6,543,307 SA processes (all regimes, including self-employed workers) from the year 2019, of whom 5,814,494 had a SA process for common eventualities and 728,813 for professional eventualities; all these SA processes were extracted from the databases of the Social Security system. Of the total sample, 685,752 cases corresponded to the databases of the Medical Units of the INSS (583,118 employees and 102,624 self-employed workers), who were those who underwent a medical inspection check.

The unit of analysis was the Spanish Autonomous Community.

The dependent variable is workers who have started and/or finished a period of SA in 2019.

The independent variables were:Age: workers aged between 16 and 65 years and older, divided into age ranges: 16–25, 26–35, 36–45, 46–55, 56–65 and >65 years.Gender: men and women contributing to the Social Security system.Occupational level: based on the International Standard Classification of Occupations (ISCO-08), developed by the International Labor Organization [21]. Armed forces occupations (Group 10) were excluded, since the SA in these workers is not managed or controlled by the Public Health Service or the INSS. For this study, we used the ISCO in level 1 aggregation, that is, the one that includes the large occupational groups (Group 1 to Group 9).Diagnostic episode of the International Classification of Diseases (ICD-10), which has been active since January 2016 in Spain.

### 2.2. Measurement of Sickness Absence

The indicators used regarding SA were:SA rate: number of workers in SA per 1000 affiliated workers, according to gender, age ranges, AC, occupational level and activity of the company.Average duration: refers to the duration, in days, of the SA processes that ended in 2019. It was calculated by dividing the total number of days in SA of those processes that started and/or finished a period of SA in 2019 (regardless of its end date) by the total number of SA processes.Number of SA cases in 2019: means the number of SA processes analysed in the INSS Medical Units in 2019. It shows the SA frequency.

### 2.3. Statistical Methods

The outcomes of the study were the SA rates, i.e., number of SA cases of average duration SA episode in the study population during 2019. Since both incidence and length of sickness absence are different among men and women, and there are clear differences in the association of occupational class with sickness absence by gender [10], the analyses were performed separately for men and women and also by age.

Descriptive statistics were obtained for all variables on the number of SA cases and the number of absence days for each AC.

Due to the importance of long-term SA, a descriptive study was performed to analyze the following parameters related to SA average duration: central tendency measures (average, median and mode), dispersion measures (standard deviation, variance of the sample and range) and position measures (kurtosis and skewness), with a confidence interval of 95% of the average. For the processing of the data, we used the descriptive statistical program of Excel^®^ 2010 (Microsoft office Excel 2010, USA).

## 3. Results

The total sample analysed constituted 6,543,307 workers in SA from the year 2019, distributed in the different ACs of Spain. The reason for selecting 2019 over 2020 is that the latter was an atypical year due to the coronavirus pandemic.

The data of some ACs must be interpreted with caution, since there is a high per-centage of cases in which the profession of the worker was not documented, as is the case of Aragon (90%), Extremadura (26%), Navarre (94%) and the Basque Country (90%). In the rest of the ACs, the percentage of non-informed occupations ranges between 3% and 6%, thus these data are more reliable. Despite this limitation, the coverage of this study overcomes this issue.

The results of the descriptive analysis according to ISCO-08 in relation to SA duration in the different ACs are available online (Appendix A). The data are highly scattered in all occupations, with “Professional” showing the most homogeneous mean durations in all ACs. The group “Managers” shows the longest mean duration (n = 69.17) and the group “Professional” presents the shortest mean duration (n = 41.96). Moreover, the group “Managers” shows the highest range of variation between mean durations (n = 66.69).

As is shown in Table 1, the groups “Elementary occupations”, “Craft and related trades workers” and “Clerical support workers” present the highest SA rates in almost all the ACs. On the other hand, the group “Managers” show the lowest SA rates.

Besides, the high SA rates in Catalonia are noteworthy, particularly in the groups “Clerical support workers” (n = 49.62), “Services or sales workers” (n = 48.11) and “Elementary occupations” (n = 59.60). La Rioja also stands out in the group “Craft and related trades workers” (n = 56) and Madrid, in the groups “Clerical support workers” (n = 55.08) and “Skilled, agricultural forestry and fishery workers” (n = 54.78).

Table 2 gathers the mean durations (average number of days) by gender and ISCO. The longest duration corresponds to women of the group “Skilled agricultural, forestry and fishery workers” and men of the group “Managers” in almost all ACs, although there is great variability between ACs.

In many occupational groups, there are very few differences in the mean duration of SA between men and women, as is the case of the group “Technicians and associate professionals” in Andalusia, “Service and sales workers” in Asturias, “Clerical support workers” in Andalusia, “Service and sales workers” in Aragon and Balearic Islands, and “Plant and machine operators, and assemblers” in Madrid.

The greatest difference in SA duration by gender in almost all CAs corresponds to the group “Elementary occupations”, with a much longer mean duration in women, as is the case in Galicia (men: 64 days; women: 109 days), Asturias (men: 54 days; women: 90 days) and Castile and Leon (men: 47 days; women: 76 days).

The ACs with lower gender differences in mean SA duration by occupational group are Madrid, Navarre and the Basque Country, which, in addition, are among the ACs with the shortest duration in all levels (ISCO-08).

As can be observed, the mean durations of all the occupations increase with the age of the worker in all ACs, as is shown in Table 3, except in the group “Elementary occupations”, where the mean duration of SA in people aged over 65 years is shorter than in younger age ranges.

Table 3, Table 4 and Table 5, show that, in all the age ranges, the shortest mean durations are found in the most elemental occupations (i.e., in the lowest hierarchies), and the duration increases in all age ranges with the increasing occupational hierarchy, except in the group “Professionals”, who have shorter means than other groups of lower occupational hierarchy or lower qualification level.

The shortest mean SA durations are found in the age range 16–25 years in the group “Plant and machine operators, and assemblers”, whereas the longest mean SA durations are found in the group “Managers” in the age range over 65 years.

Interestingly, in the age range 16–25 years the mean duration is longer than in other age ranges in the case of “Service and sales workers”, being longer than that of the next two age ranges (26–35 and 36–45) in almost all ACs.

Data relating to the number of SA cases that finished in 2019, analyzing the different diagnostic episodes (ICD-10) in relation to occupational level following the ISCO-08, are available online (Appendix A).

There is a high percentage of cases in which the occupation was coded, which may distort the data.

Among the informed cases, the largest number of workers in SA is concentrated in the episode “Diseases of the musculoskeletal system and connective tissue”, in the occupational groups “Elementary occupations” (n = 28,626), “Service and sales workers”, (n = 26,448) and “Plant and machine operators, and assemblers” (n = 9267), followed by the episode “Mental and behavioral disorders”, which mainly affects the occupational groups “Service and sales workers” (n = 12,741), “Elementary occupations” (n = 7467) and “Professionals” (n = 6813).

## 4. Discussion

As we hypothesized that new measures could be introduced to improve workplaces, workers could take up their jobs sooner, and benefits could be managed more efficiently in AC, the results of this study reveal findings related to SA as a function of worker occupation in Spain, describing the situation immediately before the SARS-CoV-2 pandemic by Autonomous Community.

The problem when analyzing SA by occupational classes was the high percentage of SA cases in which the occupational class was not documented, which occurred in several specific communities: Navarre (94%), Aragon (90%), the Basque Country (90%) and Extremadura (26%). Therefore, the results in these ACs have not been reliable and may have altered the results, as was previously commented upon; however, the study is still of great interest, as it shows the distribution of SA.

Great differences were found in all occupational groups, as is reflected by the high standard deviations obtained in the statistical analysis.

The highest SA rates correspond to “Elementary occupations”, “Craft and related trades workers” and “Clerical support workers”. These data are in line with the findings of previous studies, especially regarding the most elemental occupations and those with greater physical responsibility [10,22]. The most surprising finding of this study, in contrast with the international literature, could be the high SA rate in workers with administration duties and in office workers [17]. This could be due to the fact that a high percentage of workers with these administrative occupations are in the public sector, who show higher SA rates than those in the private sector [23,24].

Regarding SA duration, the longest durations are found in women of the group “Skilled agricultural, forestry and fishery workers”, especially in Galicia, Cantabria, Castile and Leon, Asturias, Murcia and Andalusia. However, these results must be considered with caution, since the percentage of female workers in this episode is very insignificant with respect to other occupations taken by women [20]. A possible explanation for this finding is the large number of such workers who are self-employed, among whom the mean durations are usually very long [25].

With respect to the analysis by age range, the mean SA duration increases with the age of the worker, with this effect being less pronounced in “Elementary occupations”. This could be due to several reasons. On the one hand, there could be an information bias in these professions, and thus, in reality, there may be few SA processes in which the occupation and age were documented, as was previously commented upon. On the other hand, this could also be due to the natural selection of the more elemental activities, which, for their physical load, exclude workers as their age increases; in addition, the natural process is that older workers go up in the hierarchy of the company, which is why elemental occupations are carried out by younger individuals [26]. Their mean SA duration, by AC and age range, is much shorter than that of other occupational groups, and this group shows the shortest mean durations in people over 65 years of age with respect to other age ranges. The shorter duration in this group could be due to the fact that, in many cases, these are precarious, temporary, half-time jobs performed by workers with little qualification, generally with low economic resources, who fear being fired if they extend their SA without justification [4,27].

In line with what is described above, the results of the number of SA cases as a function of worker occupation and diagnostic episode show that the diagnoses with higher SA rate in all occupations are musculoskeletal disorders, followed by mental disorders [17].

The proportional distribution between groups is the most relevant aspect. Thus, “Service and sales workers”, “Elementary occupations”, “Plant and machine operators, and assemblers” show a much greater rate of musculoskeletal disorders, which, in most cases, is three times more frequent than that of mental disorders. This is in agreement with the results of previous studies on this topic, which report a greater prevalence of pathologies at the locomotor level in blue collar workers, with precarious jobs, more disadvantaged physical conditions and a more demanding physical load [18,22,28,29].

Mental disorders gain relevance in occupations with higher qualification levels, such as “Professional, Managers, and Service” and sales workers. Previous studies have also reported these findings, indicating that these occupations can cause great personal exhaustion, especially those related to healthcare and caregiving [30,31,32], whereas, in managers, SA by mental disorders is rather attributed to the high stress levels, although, in general, few directors and managers initiate a sick leave for this reason [33,34].

It is also important to highlight the proportion of neoplasms and cardiovascular diseases as the cause of AS in managers, as well as neoplasms in scientific and intellectual professionals in health and education, which indicates that in highly qualified workers with a higher education level and higher degree of responsibility, when they initiate a SA process it is generally due to the severity of the pathology that causes it [13,34], with presenteeism being very frequent in both occupational groups [35,36,37]. These data are in line with those of previous studies published internationally and in Spain [38,39,40].

The results of this study show important differences between the Autonomous Communities in Spain.

The Autonomous Communities with the highest SA rates in almost all occupational groups are Catalonia and Madrid. These two Autonomous Communities have the lowest unemployment figures in Spain, the highest salaries and the highest percentage of employed workers [20]. These results could be explained on the basis of other studies showing that employees have higher rates of SA than the self-employed, that in areas where unemployment is lower there is more SA, as the worker is not so afraid of being fired for being on sick leave, and that workers with higher salaries also have more SA [13,27,41,42,43].

The highest average duration is found in Galicia, Extremadura and Castile and Leon. These three Autonomous Communities have an older population, with a higher percentage of self-employed workers and most of their workers are agricultural workers with precarious jobs, which means that sick leave is longer, in line with other studies [4,26,27].

However, we must be careful with these statements, as the descriptive design of the study does not allow us to draw conclusions about causality, and there may be more causes than those described here that explain these findings. It would be interesting to carry out further studies, with a different statistical design, to explore this subject in more detail.

Another important limitation of this study is that only one year is analysed (2019), so we lack a time perspective that would allow us to assess the trend in SA in Spain, in relation to our study hypothesis.

## 5. Conclusions

The highest SA frequency is found among workers of the lowest occupational classes. The longest mean duration corresponds to workers with higher qualification, especially in the group “Managers”. In Spain, there is a high SA rate in “Clerical support workers”, which is in contrast with the findings of studies conducted in other European countries. In blue collar workers, there is a predominance of SA by musculoskeletal disorders. SA by mental disorders acquires relevance in the occupational groups with higher qualification levels.

## Figures and Tables

**Table 1 ijerph-18-11453-t001:** Sickness absences rates according to occupational group (ISCO-08) and Autonomous Communities (2019).

	Managers	Professional	Technicians and Associate Professionals	Clerical Support Workers	Services and Sales Workers	Skilled Agricultural, Forestry and Fishery Workers	Craft and Related Trade Workers	Plant and Machine Operators and Assemblers	Elementary Occupations
**Andalusia**	10.40	15.04	27.09	28.34	24.09	15.35	25.05	23.11	30.17
**Aragon**	0.48	0.60	1.29	0.89	2.39	1.21	6.48	3.18	6.59
**Asturias**	10.70	13.27	17.39	24.79	22.42	11.64	25.91	26.36	32.68
**Balearic Islands**	7.29	17.02	19.30	25.70	32.06	9.64	22.51	22.45	42.01
**Canary Islands**	8.18	24.55	24.76	31.69	28.76	14.80	26.73	23.94	34.79
**Cantabria**	9.51	22.21	17.70	30.64	23.12	14.57	27.64	25.11	30.09
**Castile and Leon**	12.11	11.92	18.72	22.03	18.46	6.56	29.49	20.86	33.22
**Castile La Mancha**	9.38	12.40	16.13	26.02	19.32	6.37	24.41	19.46	36.12
**Catalonia**	11.86	27.41	35.77	49.62	48.11	27.34	41.33	34.60	59.60
**Extremadura**	9.92	14.99	9.28	23.02	17.89	11.06	15.34	16.58	22.13
**Galicia**	12.43	13.73	16.63	18.17	20.91	8.15	32.09	17.59	30.70
**Madrid**	14.94	18.97	18.09	55.08	34.85	54.78	31.34	43.50	41.56
**Murcia**	15.83	17.59	23.14	25.86	28.54	17.59	31.53	21.37	26.24
**Navarre**	0.25	0.25	0.89	0.42	1.86	0.72	7.60	3.77	5.62
**La Rioja**	10.29	16.29	19.99	23.87	24.45	12.95	56.00	21.48	33.83
**Valencia**	10.93	13.47	20.69	15.67	18.02	12.65	27.69	17.39	25.11
**Basquet Country**	0.51	0.69	1.50	0.72	2.98	2.36	9.29	3.98	8.79
**Ceuta**	17.65	15.42	12.40	32.37	18.04	10.83	32.34	15.24	37.20
**Melilla**	7.23	16.88	29.52	28.74	22.84	11.00	27.96	35.48	42.09

**Table 2 ijerph-18-11453-t002:** Average duration of sickness absence by gender and ISCO-08.

	Managers		Professional		Technicians and Associate Professionals		Clerical Support Workers		Services and Sales Workers		Skilled Agricultural, Forestry and Fishery Workers		Craft and Related Trades Workers		Plant and Machine Operators, and Assemblers		Elementary Occupations	
	Man	Woman	Man	Woman	Man	Woman	Man	Woman	Man	Woman	Man	Woman	Man	Woman	Man	Woman	Man	Woman
**Andalusia**	80	68	44	42	45	45	38	38	48	47	75	92	45	51	50	54	51	69
**Aragon**	38	36	40	48	37	45	37	46	37	37	40	34	35	33	40	38	32	37
**Asturias**	99	84	53	48	61	57	61	52	67	63	107	143	61	72	71	75	54	90
**Balearic Islands**	68	60	36	35	32	31	32	31	34	35	39	65	35	42	39	47	30	51
**Canary Islands**	82	67	35	34	44	43	41	43	44	43	60	61	44	50	53	62	40	58
**Cantabria**	83	71	39	38	53	50	31	41	59	57	115	154	54	66	59	61	49	81
**Castile and Leon**	87	86	54	57	52	55	40	44	62	64	93	134	47	51	50	52	47	76
**Castile La Mancha**	73	65	44	54	39	47	29	38	46	51	64	68	39	48	40	41	39	52
**Catalonia**	55	44	24	27	26	27	21	25	30	33	41	49	31	36	34	36	29	47
**Extremadura**	86	91	52	59	46	59	52	54	59	65	76	104	48	69	51	55	52	68
**Galicia**	105	95	54	55	63	66	57	60	67	72	119	181	63	78	74	85	64	109
**Madrid**	44	39	26	28	25	28	23	25	29	28	30	33	29	31	28	28	24	38
**Murcia**	86	92	42	43	46	52	43	44	52	54	76	102	44	69	51	63	51	74
**Navarre**	42	25	45	23	29	40	42	35	33	32	22	24	28	34	25	32	31	44
**La Rioja**	66	50	28	34	37	43	36	36	44	45	43	55	34	34	35	32	37	55
**Valencia**	83	81	49	51	48	50	38	45	54	54	62	80	45	54	45	53	46	71
**Basque Country**	34	41	39	40	37	43	32	39	31	34	32	33	29	35	31	38	31	40
**Ceuta**	72	49	43	43	54	51	56	44	46	47	54	18	47	62	61	62	49	55
**Melilla**	110	90	52	37	57	37	53	47	51	45	37	42	57	28	64	51	40	55

**Table 3 ijerph-18-11453-t003:** Average duration of sickness absence by aged ranges and ISCO-08 (I).

	Managers						Professional						Technicians and Associate Professionals					
	16–25	26–35	36–45	46–55	56–65	>65	16–25	26–35	36–45	46–55	56–65	>65	16–25	26–35	36–45	46–55	56–65	>65
**Andalusia**	36	48	57	82	109	144	18	30	38	50	61	88	21	30	40	54	76	102
**Aragon**		45	24	24	83		41	35	41	52	59	1	35	35	38	44	48	30
**Asturias**	43	54	69	96	118	114	21	32	43	60	66	80	28	35	53	71	94	111
**Balearic Islands**	27	36	46	65	107	153	16	25	32	43	57	57	11	19	30	43	65	63
**Canary Islands**	38	47	62	75	109	150	15	25	34	38	46	64	22	34	41	53	62	102
**Cantabria**	70	38	53	91	109	87	11	26	35	46	55	72	23	34	46	60	87	150
**Castile and Leon**	53	53	65	88	117	137	20	41	52	63	75	79	24	35	48	63	85	101
**Castile La Mancha**	52	45	55	76	100	123	18	41	49	61	67	83	22	30	40	53	66	110
**Catalonia**	14	26	38	56	81	138	11	19	25	33	45	56	11	18	25	35	53	86
**Extremadura**	53	77	73	86	111	142	35	47	58	61	63	59	37	42	49	59	65	18
**Galicia**	43	65	81	106	133	162	25	40	53	63	68	66	31	44	59	77	92	127
**Madrid**	16	28	36	45	59	86	11	22	28	32	37	50	13	20	26	33	43	63
**Murcia**	45	58	71	90	119	184	19	33	39	48	58	44	23	31	45	59	82	106
**Navarre**	8	6	22	41	76		14	22	32	36	58		31	28	30	24	57	230
**La Rioja**	18	34	35	56	103	144	19	24	32	37	42	67	21	26	39	48	62	115
**Valencia**	40	56	67	89	108	136	21	38	48	56	70	74	25	34	44	57	77	126
**Basque Country**	30	21	33	35	55	38	40	33	40	43	45	59	47	37	30	43	47	26
**Ceuta**	69	33	26	66	85	93	37	34	37	47	56	22	16	45	50	60	56	144
**Melilla**	9	31	72	103	207	203	15	34	35	44	51	158	56	34	44	58	57	9

**Table 4 ijerph-18-11453-t004:** Average duration of sickness absence by aged ranges and ISCO-08 (II).

	Clerical Support Workers						Service and Sales Workers						Skilled Agricultural, Forestry and Fishery Workers					
	16–25	26–35	36–45	46–55	56–65	>65	16–25	26–35	36–45	46–55	56–65	>65	16–25	26–35	36–45	46–55	56–65	>65
**Andalusia**	15	29	35	46	60	84	50	44	41	57	79	69	37	48	65	96	118	142
**Aragon**	21	30	40	55	47		47	43	58	81	100	240	16	35	43	40	42	103
**Asturias**	19	36	45	65	77	83	61	54	70	103	115	61	26	74	99	133	156	94
**Balearic Islands**	12	19	29	39	56	82	54	53	56	75	124	70	17	27	31	56	82	282
**Canary Islands**	20	31	43	45	57	80	74	82	71	76	102	86	17	36	48	68	101	77
**Cantabria**	7	26	40	54	70	88	51	54	58	76	102	93	39	59	102	145	161	108
**Castile and Leon**	12	31	43	54	70	51	49	40	46	70	89	51	36	62	74	106	133	137
**Castile La Mancha**	14	25	34	46	62	53	48	47	45	72	97	78	41	42	56	76	85	59
**Catalonia**	9	16	23	32	46	84	25	25	27	40	59	58	16	28	36	50	78	113
**Extremadura**	26	38	51	60	69	64	129	132	117	151	165	97	43	58	72	94	106	114
**Galicia**	26	44	54	66	85	111	74	64	64	78	92	66	91	87	122	158	190	228
**Madrid**	10	19	24	29	38	57	29	23	24	34	49	43	12	19	27	35	50	87
**Murcia**	19	30	41	53	71	91	67	52	49	69	87	67	51	62	72	100	131	143
**Navarre**	16	42	27	44	68		21	33	51	97	74	104	15	18	20	24	37	17
**La Rioja**	13	25	29	41	61	52	30	33	39	68	87	44	15	22	41	54	74	131
**Valencia**	17	29	40	52	69	95	53	40	36	45	60	43	31	41	55	74	98	172
**Basque Country**	21	39	31	40	45		26	37	43	68	70	80	16	44	30	32	33	
**Ceuta**	40	28	36	53	64	213	73	92	111	112	103	25	3	315	25	25	53	30
**Melilla**	34	49	43	43	70	21	71	83	92	105	169	50	14	12	28	46	67	

**Table 5 ijerph-18-11453-t005:** Average duration of sickness absence by aged ranges and ISCO-08 (III).

	Craft and Related Trades Workers						Plant and Machine Operators, and Assemblers						Elementary Occupations					
	16–25	26–35	36–45	46–55	56–65	>65	16–25	26–35	36–45	46–55	56–65	>65	16–25	26–35	36–45	46–55	56–65	>65
**Andalusia**	15	16	19	26	35	19	8	8	11	16	21	12	23	18	18	29	36	29
**Aragon**	54	62	96	99	96	50	16	22	41	53	47	51	30	28	37	42	51	6
**Asturias**	17	29	37	48	49	26	9	16	24	32	27	5	13	10	14	27	34	20
**Balearic Islands**	13	14	24	34	58	32	3	5	8	15	26	12	14	13	18	32	55	31
**Canary Islands**	9	13	17	24	31	19	4	7	12	19	25	19	17	22	22	33	42	23
**Cantabria**	14	24	34	42	61	25	9	10	17	24	28	16	11	10	14	23	33	19
**Castile and Leon**	29	28	36	45	59	23	10	13	19	25	26	10	17	13	17	32	40	22
**Castile La Mancha**	30	30	36	52	65	49	10	13	16	27	29	11	26	21	24	39	51	42
**Catalonia**	6	8	14	20	31	23	3	5	9	14	20	12	5	6	9	16	24	17
**Extremadura**	29	31	33	40	45	13	14	13	17	21	20	13	42	36	29	43	44	31
**Galicia**	44	44	53	63	84	30	10	13	18	27	28	15	15	13	17	32	49	33
**Madrid**	5	6	10	14	18	12	4	5	7	10	13	10	5	6	8	14	21	21
**Murcia**	30	27	32	42	49	19	11	11	15	25	25	16	34	26	32	41	49	24
**Navarre**	45	80	145	188	166	0	15	31	49	86	69	0	15	29	36	70	67	48
**La Rioja**	31	37	48	81	95	18	5	9	15	26	27	7	9	10	13	27	35	23
**Valencia**	18	18	23	29	35	16	10	9	13	19	22	12	16	12	14	22	29	11
**Basque Country**	36	52	81	91	90	29	10	18	30	41	37	1	16	21	26	39	35	22
**Ceuta**	15	11	23	28	27	8	7	12	17	22	28	0	15	24	31	47	44	11
**Melilla**	16	15	20	28	50	143	6	10	10	19	34	0	22	22	28	50	87	

## Data Availability

You can consult the data used in https://www.seg-social.es/wps/portal/wss/internet/EstadisticasPresupuestosEstudios/PublicacionesDocumentacion/47999/c53ff6f9-aef0-4a67-b26f-0c6031efbeb8; https://www.ine.es/ (accessed on 27 August 2021).

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
