# Peer review of "Occupation and Sickness Absence in the Different Autonomous Communities of Spain"

_ijerph, 2021, doi:10.3390/ijerph182111453_

Round 1

Reviewer 1 Report

I am unclear about your motivation for this work, other than somebody else hasn't done exactly this study before. 

Also, the charts and tables contain too much data, meaning it is hard to draw out meaningful information or understand your claims.

You should include a justification for why you present largely uncontrolled descriptive statistics. These autonomous regions are likely very different in important ways. As a result, understanding differences in sickness absences is very challenging.

Author Response

ITEMIZED LIST OF THE REVIEWERS COMMENTS

Manuscript ID: ijerph-1376558

Title: Occupation and sickness absence in the different Autonomous Communities of Spain

Dear Reviewer,

We greatly appreciate the editor´s and reviewers’ kind and encouraging comments about our study. We have followed their suggestions, trying to incorporate them into the revised version of our manuscript. We uploaded the tracked changes manuscript, the clean version revised manuscript and itemized point-by-point response to the reviewer’ comments are presented below.

Editor´s and Reviewers´ comments:

*Reviewer 1

RV: Reviewer

AA: Authors

RV:1. I am unclear about your motivation for this work, other than somebody else hasn't done exactly this study before. 

AA: First, authors want to thank the modifications suggested by the reviewer and his/her effort to improve our manuscript. Following the reviewer suggestion, we have re-written the introduction to be more sophisticated around area of sickness absence and occupation. We have eliminated a paragraph.  Several new paragraphs have been added in the introduction. Firstly, a new paragraph was added in lines 50 to 60 on page 2 in the revised clean version in the introduction section:

Like most European countries, Spain’s public sickness insurance spending is a major component of its social security system. The total financial cost of SA in consoli-dated Social Security budgets for 2019 in Spain reached 11 554 711.16 €, which is 8.89% of the general Social Security budget[4]. Besides, the importance of SA caused by disease or accident is not only determined by the social and financial costs that each State dedicates to it, but also by the resources that employers assign to it, as well as the productivity losses and the deterioration of these workers’ health . It has been reported that work absenteeism due to illness is, in itself, a risk factor to fall sick, new periods of SA, unemployment, permanent disability pension, social exclusion and death[5–8].Long SA periods generate both an early retirement from the job market, a slower salary increase and an impoverishment of household wealt[9].

Knowledge of the determinants of sickness absence and of the accumulation of sickness absence days is important to be able to target effective measures that aim to prevent both incidence and prolongation of sickness absence and to curb its harmful consequences[10]”.

Secondly, a new paragraph was added in lines 112 to 136 on page 3 in the revised clean version in the introduction section:

“Spain is divided into 19 Autonomous Community. In each region there are significant differences in education, economic situation, unemployment rate and public health system. By sex, the percentage of men and women is balanced. The number of women is slightly higher, although the number of working men is still higher. By age, the regions with the youngest population are mainly those on the Mediterranean coast and also Castile La Mancha, Extremadura and the Community of Madrid. The northwestern regions are more aged. The regions with the highest rates of active population were the Canary Islands, the Balearic Islands, the Community of Madrid and Catalonia and those with the lowest rates were the Cantabrian, Western and Central regions. Service sector was the majority of the work- force (76%) in the islands, the Community of Madrid and Andalusia. Industry was the predominant sector in Navarre, the Basque Country and La Rioja; construction in Castile La Mancha; and agriculture and farming in Murcia[20].

A descriptive study on SA was conducted in Spain in 2018, also by Autonomous Community (AC), although SA was not analysed as a function of the worker’s occupation(4). The authors concluded that further research was necessary due to the differences found in SA in the different ACs regarding age, gender, whether the worker was employed or self-employed, the cause of the sick leave (work-related or non-work-related SA) and the relationship with the activity in the company.

Therefore, the aim of the present study was to analyse the differences in SA in terms of occupational class, according to the International Standard Classification of Occupations (ISCO-08)[21], focussing on the different regions of the country and according to the main characteristics of SA and differences between AC. Thanks to this study, new measures could be introduced to improve workplaces, workers could take up their jobs sooner, SA benefits could be managed more efficiently in AC, and the negative consequences of SA could be reduced”.

RV:2. Also, the charts and tables contain too much data, meaning it is hard to draw out meaningful information or understand your claims.

AA: Thank you for your comment. We agree that tables contain too much data. Following the reviewer´s recommendation, we have improved our charts and tables in the result section of the revised manuscript.

RV:3. You should include a justification for why you present largely uncontrolled descriptive statistics. These autonomous regions are likely very different in important ways. As a result, understanding differences in sickness absences is very challenging.

AA: Thank you for your comment. We have attempted to better clarify our uncontrolled descriptive statistics in the method section. The new sentence now reads:

“This is an observational study. The statistical calculations used to refer to all the people affiliated with the system who are entitled to receive the SA benefit, calculated according to the data provided by the Public Employment Service, the General Treasury of Social Security, INSS and the Social Institute of the Navy”

2.3. Statistical Methods

The outcomes of the study were the SA rates, number of SA cases of average duration SA episode in the study population during 2019. Since both incidence and length of sickness absence are different among men and women, and there are clear differences in the association of occupational class with sickness absence by gender(10), the analyses were performed separately for men and women and also by age.

Descriptive statistics were obtained for all variables on the number SA cases and the number of absence days for each AC.

Due to the importance of long-term SA, a descriptive study was performed to analyse the following parameters related to SA average duration: central tendency measures (average,…

Please, do not hesitate to contact me, if you require further corrections and information.

Thank you in advance

Reviewer 2 Report

1.The introduction is no clear storyline: too many topics wanted to be cover in a no-systematic order. Methodology clearness in achieving the aim of the study is lacking.

2.You need to go beyond describing a series of relevant references, and tell us how your interpretation of the literature shows the gaps that exist, and how the proposed approach to the literature brings about novel opportunities to reinterpret the literature that will allow an advancement in our understanding in the field. This feels mechanical and lacking in originality. Additionally, more updated literature could have been used.

3.The description of the Methodology is lacking in detail. The analysis of the data and the methodology is not shown as robust from the scientific point of view and is not well explained. The paper need a deeper level of analysis of the key issues. How do you explain your statistical analysis outcomes and how do they contribute the current research?

4.The discussion and conclusions do not provide enough detail. The discussion needs to be more than sequencing data and some related references, it needs to be a coherent and cohesive set of arguments that take us beyond this study in particular.

Author Response

ITEMIZED LIST OF THE REVIEWERS COMMENTS

Manuscript ID: ijerph-1376558

Title: Occupation and sickness absence in the different Autonomous Communities of Spain

Dear Reviewer,

We greatly appreciate the editor´s and reviewers’ kind and encouraging comments about our study. We have followed their suggestions, trying to incorporate them into the revised version of our manuscript. We uploaded the tracked changes manuscript, the clean version revised manuscript and itemized point-by-point response to the reviewer’ comments are presented below.

Editor´s and Reviewers´ comments:

*Reviewer 2

RV: Reviewer

AA: Authors

RV: 1. The introduction is no clear storyline: too many topics wanted to be cover in a no-systematic order. Methodology clearness in achieving the aim of the study is lacking.

AA: Thank you for your comment. We have attempted to better clarify our introduction and purpose statement. Following the reviewer suggestion, we have re-written the introduction to be more sophisticated around area of sickness absence and occupation. A new paragraph was added in lines 50 to 60 on page 2 in the revised clean version in the introduction section:

Like most European countries, Spain’s public sickness insurance spending is a major component of its social security system. The total financial cost of SA in consoli-dated Social Security budgets for 2019 in Spain reached 11 554 711.16 €, which is 8.89% of the general Social Security budget[4]. Besides, the importance of SA caused by disease or accident is not only determined by the social and financial costs that each State dedicates to it, but also by the resources that employers assign to it, as well as the productivity losses and the deterioration of these workers’ health . It has been reported that work absenteeism due to illness is, in itself, a risk factor to fall sick, new periods of SA, unemployment, permanent disability pension, social exclusion and death[5–8].Long SA periods generate both an early retirement from the job market, a slower salary increase and an impoverishment of household wealt[9].

Knowledge of the determinants of sickness absence and of the accumulation of sickness absence days is important to be able to target effective measures that aim to prevent both incidence and prolongation of sickness absence and to curb its harmful consequences[10]”.

Secondly, a new paragraph was added in lines 112 to 136 on page 3 in the revised clean version in the introduction section:

“Spain is divided into 19 Autonomous Community. In each region there are significant differences in education, economic situation, unemployment rate and public health system. By sex, the percentage of men and women is balanced. The number of women is slightly higher, although the number of working men is still higher. By age, the regions with the youngest population are mainly those on the Mediterranean coast and also Castile La Mancha, Extremadura and the Community of Madrid. The northwestern regions are more aged. The regions with the highest rates of active population were the Canary Islands, the Balearic Islands, the Community of Madrid and Catalonia and those with the lowest rates were the Cantabrian, Western and Central regions. Service sector was the majority of the work- force (76%) in the islands, the Community of Madrid and Andalusia. Industry was the predominant sector in Navarre, the Basque Country and La Rioja; construction in Castile La Mancha; and agriculture and farming in Murcia[20].

A descriptive study on SA was conducted in Spain in 2018, also by Autonomous Community (AC), although SA was not analysed as a function of the worker’s occupation(4). The authors concluded that further research was necessary due to the differences found in SA in the different ACs regarding age, gender, whether the worker was employed or self-employed, the cause of the sick leave (work-related or non-work-related SA) and the relationship with the activity in the company.

Therefore, the aim of the present study was to analyse the differences in SA in terms of occupational class, according to the International Standard Classification of Occupations (ISCO-08)[21], focussing on the different regions of the country and according to the main characteristics of SA and differences between AC. Thanks to this study, new measures could be introduced to improve workplaces, workers could take up their jobs sooner, SA benefits could be managed more efficiently in AC, and the negative consequences of SA could be reduced”.

RV: 2. You need to go beyond describing a series of relevant references, and tell us how your interpretation of the literature shows the gaps that exist, and how the proposed approach to the literature brings about novel opportunities to reinterpret the literature that will allow an advancement in our understanding in the field. This feels mechanical and lacking in originality. Additionally, more updated literature could have been used.

AA: First, authors want to thank the modifications suggested by the reviewer. Totally agree with you. Regarding recommendation the introduction was rewritten to better understand in line 50 to 60 on page 2. Without a doubt, their recommendations help us to give more clarity and structure to the introduction part.

RV: 3. The description of the Methodology is lacking in detail. The analysis of the data and the methodology is not shown as robust from the scientific point of view and is not well explained. The paper need a deeper level of analysis of the key issues. How do you explain your statistical analysis outcomes and how do they contribute the current research?

AA: Following the reviewer’s recommendation, we have improved the methodology part. Thank you for your comment.

RV: 4. The discussion and conclusions do not provide enough detail. The discussion needs to be more than sequencing data and some related references, it needs to be a coherent and cohesive set of arguments that take us beyond this study in particular.

AA: Thank you for your comment. We have attempted to better clarify our discussion and purpose statement. Following the reviewer suggestion, we have re-written the discussion. A new paragraph was added in lines 388 to 410  on page 18 in the revised clean version in the discussion section:

“The results of this study show important differences between the Autonomous Communities in Spain.

The Autonomous Communities with the highest SA rates in almost all occupational groups are Catalonia and Madrid. These two Autonomous Communities have the lowest unemployment figures in Spain, the highest salaries and the highest percentage of employed workers[20]. These results could be explained on the basis of other studies showing that employees have higher rates of SA than the self-employed, that in areas where unemployment is lower there is more SA, as the worker is not so afraid of being fired for being on sick leave, and that workers with higher salaries also have more SA[13,27,41–43].

The highest average duration is found in Galicia, Extremadura and Castile and Leon. These three Autonomous Communities have an older population, with a higher percentage of self-employed workers and most of their workers are agricultural workers with precarious jobs, which means that sick leave is longer, in line with other studies[4,26,27].

However, we must be careful with these statements, as the descriptive design of the study does not allow us to draw conclusions about causality, and there may be more causes than those described here that explain these findings. Further studies it would be interesting to carry out, with a different statistical design, to explore this subject in more detail.

Another important limitation of this study is that only one year is analysed (2019), so we lack a time perspective that would allow us to assess the trend in SA in Spain, in relation to our study hypothesis”.

Please, do not hesitate to contact me, if you require further corrections and information.

Thank you in advance

Reviewer 3 Report

This is a very interesting article on occupational health that provides valuable information on the sociodemographic variables that affect morbidity in Spain.

The authors have developed a cross-sectional study conducted to analyse the differences in sickness abcense (SA) in terms of occupational class, according to the International Standard Classification of Occupations (ISCO-08). This study was conducted by Autonomous Community, to analyse the possible differences among them.

Abstract. It is correct.

Keywords. I would encourage authors to include more keywords such as "occupational class"

Background. A total of 21 references were used in the background and of these, 10 references (47.6%) are less than five years old. I recommend translating the Spanish references.

Material and methods.  A population study was carried out using the databases of the INSS and the Spanish Statistics  Institute (INE) including 6,543,307 SA processes (all regimes, including self-employed workers) from the year 2019.

I recommend inserting a new subsection "Statistical analysis" were to include all the information of statistical data analysis.

Results. Please, the image quality in table 2 needs to be improved. I recommend editing the table in word.

Discussion. I recommend including “limitations of the study” in the Discussion section. This research has many interesting limitations that should be addressed

Author Response

ITEMIZED LIST OF THE REVIEWERS COMMENTS

Manuscript ID: ijerph-1376558

Title: Occupation and sickness absence in the different Autonomous Communities of Spain

Dear Reviewer,

We greatly appreciate the editor´s and reviewers’ kind and encouraging comments about our study. We have followed their suggestions, trying to incorporate them into the revised version of our manuscript. We uploaded the tracked changes manuscript, the clean version revised manuscript and itemized point-by-point response to the reviewer’ comments are presented below.

Editor´s and Reviewers´ comments:

*Reviewer 3

RV: Reviewer

AA: Authors

RV: GENERAL COMMENTS:

This is a very interesting article on occupational health that provides valuable information on the sociodemographic variables that affect morbidity in Spain.

The authors have developed a cross-sectional study conducted to analyse the differences in sickness abcense (SA) in terms of occupational class, according to the International Standard Classification of Occupations (ISCO-08). This study was conducted by Autonomous Community, to analyse the possible differences among them.

RV: Minor Comments

RV: 1. Abstract. It is correct.

AA: Thank you for your comment

RV: 2. Keywords. I would encourage authors to include more keywords such as "occupational class"

AA: Thank you for your comment. A new keywords was added.

RV: 3. Background. A total of 21 references were used in the background and of these, 10 references (47.6%) are less than five years old. I recommend translating the Spanish references.

AA: Thank you for your comment. We have updated some bibliographical references.  It is not possible to translate the Spanish references into English.

RV: 4. Material and methods.  A population study was carried out using the databases of the INSS and the Spanish Statistics Institute (INE) including 6,543,307 SA processes (all regimes, including self-employed workers) from the year 2019.

AA: Thank you for your comment. We have modified this section in the revised manuscript.

RV: 5. I recommend inserting a new subsection "Statistical analysis" were to include all the information of statistical data analysis.

AA: Thank you for your comment. A new subsection was added in the revised manuscript.

RV: 6. Results. Please, the image quality in table 2 needs to be improved. I recommend editing the table in word.

AA: Thank you for your comment. We have modified table 2.

RV: 7. Discussion. I recommend including “limitations of the study” in the Discussion section. This research has many interesting limitations that should be addressed

AA: Thank you. Following the reviewer suggestion, we have included “limitations of the study”. New paragraphs were added in lines 403  to 408  on page 18 in the revised clean version in the discussion section.

Please, do not hesitate to contact me, if you require further corrections and information.

Thank you in advance

Round 2

Reviewer 2 Report

Thank you very much for the detailed and well-explained revision of the document. I think it will be a good contribution to our Journal and I recommend to accept the paper.

Author Response

ITEMIZED LIST OF THE REVIEWERS COMMENTS

Manuscript ID: ijerph-1376558

Title: Occupation and sickness absence in the different Autonomous Communities of Spain

Dear Reviewer,

We greatly appreciate the editor´s and reviewers’ kind and encouraging comments about our study. We have followed their suggestions, trying to incorporate them into the revised version of our manuscript. We uploaded the tracked changes manuscript, the clean version revised manuscript and itemized point-by-point response to the reviewer’ comments are presented below.

Editor´s and Reviewers´ comments:

*Reviewer 2

RV: Reviewer

AA: Authors

RV: 1. Thank you very much for the detailed and well-explained revision of the document. I think it will be a good contribution to our Journal and I recommend to accept the paper.

AA: Thank you for your comment. Without a doubt, your recommendations help us to give more clarity and structure to our manuscript.

Thank you very much

All the best
